# A New Medical Evaluation for Gastric Cancer Patients to Increase the Success Rate of Immunotherapy: A 2024 Update

**DOI:** 10.3390/ph17091121

**Published:** 2024-08-24

**Authors:** Gabriel Samasca, Claudia Burz, Irena Pintea, Adriana Muntean, Diana Deleanu, Iulia Lupan, Vasile Bintintan

**Affiliations:** 1Department of Immunology, Iuliu Hatieganu University of Medicine and Pharmacy, 400006 Cluj-Napoca, Romania; cristina.burz@umfcluj.ro (C.B.); nedelea@umfcluj.ro (I.P.); adriana.muntean@umfcluj.ro (A.M.); deleanu@umfcluj.ro (D.D.); 2Institute of Oncology, “Prof. Ion Chiricuta”, 400015 Cluj-Napoca, Romania; 3Department of Molecular Biology, Babes-Bolyai University, 400084 Cluj-Napoca, Romania; iulia.lupan@ubbcluj.ro; 4Department of Surgery 1, Iuliu Hatieganu University of Medicine and Pharmacy, 400006 Cluj-Napoca, Romania; vasile.bintintan@umfcluj.ro

**Keywords:** gastric cancer, immunotherapy, update, 2024, new evaluation

## Abstract

Researchers have performed numerous studies on immunotherapy because of the high death rate associated with gastric cancer (GC). GC immunotherapy research has made tremendous progress, and we wanted to provide an update on this topic. On the basis of this update, we suggest performing a new medical evaluation before initiating immunotherapy in patients with GC to increase the success rate of immunotherapies. We propose that before patients start GC immunotherapy, they should be evaluated and given a score of one to two points for the following factors: immunopathological features, molecular and genomic features, potential consequences for bacterial pathogens, potential immunotherapeutic resistance and hyperprogressive illness, and the potential to use biomarkers to gauge their prognosis and immunotherapy responses to optimize immunotherapy following surgery. The proposed scoring system could also help in the diagnosis of GC. With all the advances in genetics, immunology, and microbiology, the diagnosis of GC could be improved, not changed. Currently, patients diagnosed with GC undergo surgical resection as the only permanent solution. Patients who meet the maximum score from the presented proposal could be eligible immediately after diagnosis for immunotherapy. Therefore, immunotherapy could be a first-line option for clinicians.

## 1. Introduction

Little is known about the complex etiology of gastric cancer (GC). Patients with GC still have a poor prognosis [1]. There are still certain drawbacks to the current treatment options, which primarily involve surgical techniques (open versus laparoscopic) and chemotherapy (which has serious side effects). Moreover, the outlook for advanced GC regarding the best strategy is still concerning since there is no standard. Through its mechanism of action, immunotherapy has great potential for GC treatment [2]. Numerous immunotherapies have covered further indications and earlier therapeutic lines since the Food and Drug Administration first approved pembrolizumab in the KEYNOTE-059 clinical trial 2017 [3]. Immunotherapy is not the first step in GC treatment. However, the efficacy of immunotherapy combined with other different therapies (chemotherapy, surgery, and targeted therapies) is continuously improving [4]. However, how effective is immunotherapy alone in treating gastric cancer? (Figure 1).

This article provides an update on the major developments in GC immunotherapy research. On the basis of this update, we propose performing a new medical evaluation before starting immunotherapy for GC. We utilized the keywords “immunotherapy, gastric cancer” in the PubMed database to locate the most significant and recent publications. As previously noted, only articles from 2023 to 2024 met the inclusion requirements. Since case report articles had nothing to do with our topic, they were excluded from the research.

## 2. GC Immunotherapy Research

Novel immunotherapy approaches, such as immune checkpoint inhibitors (ICIs), tumor vaccines, adoptive immunotherapy, and nonspecific immunomodulators, hold great potential, particularly for incurable or metastatic conditions. Since the discovery of ICIs, the discipline of immunotherapy has advanced significantly [5]. ICIs have been approved to treat a wide range of malignancies, and the list of approved medications is continually expanding. These malignancies include malignant melanoma, renal cell carcinoma, non-small-cell lung cancer, head and neck cancer, Hodgkin’s disease, small-cell lung cancer, GC, esophageal cancer, breast cancer, uterine cancer, and hepatocellular carcinoma [6]. The interferon (INF)-γ response, INF-α response, glycolysis, and reactive oxygen species pathway gene sets were enriched in the protein patched homologue 1 (PTCH1-MUT) group according to gene set enrichment analysis. Thus, PTCH1 could be a useful biomarker for predicting the response to ICIs in patients with GC [7].

There is ongoing debate regarding the optimal first-line treatment for advanced GC. As a first-line treatment for advanced GC, apatinib, in conjunction with chemotherapy and immunotherapy, has good antitumour efficacy and is well tolerated by patients [8]. In second-line therapy for advanced GC, apatinib plus toripalimab treatment (anti-PD-1 therapy) demonstrated tolerable toxicity but did not improve clinical outcomes in comparison with the physician’s chosen chemotherapeutic treatment plan. Six patients (24.0%) experienced adverse events of ≥ grade 3 in arm A (patients with advanced GC who progressed after first-line chemotherapy were enrolled and received 250 mg of apatinib per day plus 240 mg of toripalimab on day 1 every three weeks), whereas nine patients (34.6%) experienced adverse events of ≥ grade 3 in arm B (the physician’s choice of chemotherapy) [9]. When anti-programmed cell death 1 (PD-1) inhibitors and chemotherapy are used to treat advanced gastric or gastroesophageal junction (G/GEJ) adenocarcinomas, patients with a programmed death ligand 1 (PD-L1) combined positive score of >5 have a considerably better response and longer progression-free survival (PFS). Immunotherapy plus chemotherapy may also be beneficial for certain subgroups within the low-PD-L1-expressing population, such as those with non-diffuse-type tumors and no peritoneal metastases [10].

Research has been conducted on the technical aspects of the T-cell response during combination immunotherapy with nivolumab and radiation. A subgroup of patients with GC may benefit from the combination of radiotherapy and nivolumab, as oligofractionated irradiation can modulate the immune system and transmit antigens [11]. In terms of the microenvironment, increased CD8+ T-cell infiltration and the formation of an immune hub involving tumor-reactive chemokine (C-X-C motif) ligand 13 (CXCL13) T-cell programs and epithelial interferon-stimulated gene programs were observed following pembrolizumab treatment. The number of patients who could benefit more from PD-1 treatments could be increased by strategies to promote increases in antitumour immune hub development [12]. Therapy based on PD-1 inhibitors has shown encouraging outcomes in the treatment of metastatic GC. Following treatment, two patients (1.9%) achieved a full response; the overall response rate (ORR) was 30.5%, and the disease control rate was 89.5%. The median overall survival (OS) was 22.0 months, while the median progression-free survival (PFS) was 9.0 months. Furthermore, in patients with human epidermal growth factor receptor 2 (HER2)-positive MGC, normal baseline levels of carcinoembryonic antigen (CEA) and the combination of PD-1 inhibitors with chemotherapy and trastuzumab independently predict prolonged PFS and OS [13].

HER2-positive cancers may greatly benefit from combination immunotherapeutic approaches employing PD-1/PD-L1 inhibitors. However, more investigations and clinical tests are necessary to clarify the best possible treatment plans that optimize therapeutic outcomes while reducing side effects [14]. In addition to trastuzumab, trastuzumab deruxtecan (DS-82101), disitamab vedotin (RC48), and other new anti-HER2 treatments are being tested in preclinical and clinical settings. Currently, the first-line treatment for this disease subtype involves immunotherapy in conjunction with anti-HER2 drugs [15]. For a long time, HER-2 was the only target in GC. Claudin-18 isoform 2 (CLDN18.2) is now known to be expressed in GC. CLDN18.2 expression in gastric malignancies indicates a “gastric differentiation” program. It is implicated in gastric differentiation through the maintenance of epithelial barrier function and the coordination of signaling pathways. Thus, the first “targeted” treatment specifically for GC is targeting CLDN18.2 [16]. According to reports, CLDN18.2 is expressed in 14–87% of gastric adenocarcinoma (GAC) patients. Because it is expressed on the outer cell membrane, CLDN18.2 can be bound by monoclonal antibodies (mAbs) [17]. There are currently several significant trials investigating the use of perioperative immunotherapy for patients with resectable stomach or gastroesophageal junction tumors. The DANTE trial intermediate analysis and the final results of the GERCOR NEONIPIGA study were published in 2022. The KEYNOTE-859 and SPOTLIGHT trials, which use ICIs and the monoclonal antibody zolbetuximab to target CLDN18.2, respectively, address an unmet need for additional targeted therapies for patients with previously untreated, HER2-negative, unresectable, or metastatic GC [18]. The first-line treatment of HER2-negative GCs has demonstrated the effectiveness of zolbetuximab and many anti-PD-1/PD-L1 inhibitors. In the first-line treatment of HER2-negative, inoperable, or metastatic GCs, a meta-analysis revealed no significant difference in PFS, OS, or ORR between various checkpoint inhibitors or between immunotherapy and anti-CLDN18.2-targeted therapies [19]. Cancer immunotherapy has completely changed as a result of ICIs. However, why is immunotherapy not always effective? Numerous factors, including those closely related to the patient’s health and lifestyle, such as genetic factors, factors related to immune system cells or the GC microenvironment, factors emerging from host cells, advanced age, biological sex, diet, hormones, pre-existing comorbidities, or even the makeup of the gut microbiome, may contribute to immunoresistance and affect the patient’s response to immune therapy. The complex tumor microenvironment (TME) of GC causes differences in the epidemiological characteristics, clinicopathological characteristics, biological behavior, treatment modalities, and pharmaceutical decisions of GC patients between Eastern and Western populations [20].

Recent developments in immunotherapy for advanced GC include chimeric antigen receptor (CAR) T-cell treatment, ICIs, cancer vaccines, and vascular endothelial growth factor A inhibitors. However, these therapeutic approaches are not without difficulties [21]. CAR-T-cell therapy, a new hematologic treatment, has been identified as a therapeutic option for CLDN18.2-positive GC. Not only was a partial response observed, but circulating tumor DNA (ctDNA) was also validated as a liquid tumor biomarker since its levels were not detected after treatment [22]. Compared with conventional CAR-T cells, CAR-T cells generated with interleukin (IL)-15/IL-15Rα presented markedly better CAR-T-cell growth, cytokine production, and cytotoxicity, as well as superior tumor control, in patients with GC. GLI pathogenesis-related 1 (GLIPR1) was upregulated following CAR-T-cell therapy, and patients with GC who expressed high levels of GLIPR1 presented lower survival rates. Conventional CAR-T cells, but not CAR-T cells, were less cytotoxic when GLIPR1 was overexpressed. The combination of GLIPR1 knockdown with CAR-T-cell therapy enhanced antitumour activity both in vivo and in vitro [23].

Tumor immunotherapy can be approached through four main strategies: adoptive immunotherapy, nonspecific immunomodulators, tumor vaccines, and ICIs. Among these immunotherapies, ICIs are the most developed and widely used cancer immunotherapy for GC. Immune-related adverse events (irAEs), such as dermatitis, diarrhea, colitis, endocrinopathy, hepatotoxicity, neuropathy, and pneumonitis, are linked to these treatments and are frequently mild but may be fatal. Regardless of the site of origin, patients with cancers characterized by high microsatellite instability (MSI) or low mismatch repair exhibit a promising response to ICIs. Therefore, identifying MSI before ICIs are used to treat GC is critical [24]. A highly conserved DNA repair mechanism that safeguards genome integrity during replication is the mismatch repair (MMR) system. Deficient MMR (dMMR) leads to a greater accumulation of genetic errors in microsatellite sequences, which causes the microsatellite instability high (MSI-H) phenotype. dMMR/MSI-H status is a valid prognostic biomarker for ICI treatment due to the high neoantigen burden, presence of tumor-infiltrating lymphocytes, and activation of PD-L1 [25]. There is a pressing need for liquid biomarkers to predict the occurrence of irAEs. Anti-PD-L1-mediated lysosomal degradation results in the production of soluble PD-L1, and this signal can be utilized to predict when irAEs will occur while patients are receiving anti-PD-L1 medication [26]. Tislelizumab, an anti-programmed cell death protein 1 antibody, was assessed as a tissue-independent monotherapy for patients with dMMR and MSI-H malignancies in the open-label, phase II RATIONALE-209 study. The participants were adults with locally advanced MSI-H/dMMR solid tumors that were either metastatic or not resectable after prior treatment. Every three weeks, patients received 200 mg of tislelizumab intravenously. An independent review committee evaluated PFS, duration of response, ORR, and the primary endpoint (Response Evaluation Criteria in Solid Tumors v1.1). The conclusion was that tislelizumab is usually well tolerated and significantly improves the ORR in patients with previously treated MSI-H/dMMR cancers that had locally progressed, were unresectable, or were metastatic [27].

The development and improvement of bispecific antibody (BsAb) structures, as well as their combination with other therapeutic modalities to increase their efficacy and overcome resistance, are the main topics of current research. What research has been carried out? Using patient-derived xenografts and organoids, a recombinant fully human IgG1 bispecific antibody, IBI315, was created, and its antitumour efficacy, as well as the underlying mechanism, were studied in vitro and in vivo in mouse tumor models reconstituted with human immune cells. Tumor immune cell recruitment and activation are also induced by IBI315 therapy. Tumor cells undergo pyroptosis mediated by gasdermin B (GSDMB), which is triggered by IBI315 and results in T-cell recruitment and activation. [28]. A new CLDN18, known by other names such as TJ-CD4B or TJ033721, the 2×4-1BB bispecific antibody ‘givastomig’, or ‘ABL111’, was created to trigger 4-1BB signaling in a way that was dependent on the CLDN18.2 interaction. In patients with GC (n = 60), 4-1BB+ T cells and CLDN18.2+ tumor cells were found to coexist. Givastomig/ABL111 could elicit only 4-1BB activation in vitro in the context of CLDN18.2 binding and bind with high affinity to cell lines expressing different amounts of CLDN18.2. There was a strong correlation between the degree of T-cell activation induced by givastomig/ABL111 therapy and the CLDN18.2 expression level in tumor cells obtained from a GC patient xenograft model [29]. Thus, tumor models were used in the current BsAb investigation. What are the future possibilities? Cancer immunotherapy based on BsAbs has opened new treatment options for many types of cancer. In the future, there is hope that the current investigation of BsAb-based immunotherapy for solid tumors may provide promising clinical application prospects [30].

Table 1 presents a summary of major developments in GC immunotherapy.

## 3. Immunotherapy Characteristics

### 3.1. Immunopathological Characteristics

Immunotherapy offers patients with metastatic GC more therapeutic options. However, as a new therapy, the predictive value of PD-L1 as a first-line treatment remains questionable, as some patients do not respond. Liver metastases may lessen a patient’s response to immunotherapy. An important immunosuppressive function in the GC metastatic microenvironment is the interaction between clusters of differentiated (CD) 8+ fatigued T lymphocytes and tumor-associated macrophages (TAMs) + secreted phosphoprotein 1 (SPP1) [31]. The risk score and infiltration abundance of CD8+ T cells were negatively correlated, as demonstrated by immune infiltration analysis. Patients with high risk scores are not responsive to anticancer immunotherapy, according to a cohort analysis of immunotherapy [32].

The presence of peripheral CD4+ T-cell subpopulations has good predictive value for treatment response and longer survival in patients with advanced GC. Subpopulations of CD4+ T cells may allow for the identification and screening of beneficial populations in patients with advanced GC [33]. Flow cytometric analysis of xenograft GC tumors revealed that M2 macrophages (alternatively activated macrophages), myeloid-derived suppressor cells (MDSCs), and regulatory T cells (Tregs) were less common in tumors in which C-type lectin domain family 11 member A (CLEC11A) was suppressed. On the other hand, cytotoxic CD8+ T cells and helper CD4+ T cells are more prevalent [34]. The antitumour functions of TAMs, tumor-associated neutrophils (TANs), and various stromal cells, such as cancer-associated fibroblasts (CAFs), endothelial cells (ECs), stellate cells, and mesenchymal stem/stromal cells (MSCs), can be inhibited by epigallocatechin gallate (EGCG), which can also attenuate the immunosuppression of these cells, Tregs, and MDSCs. EGCG can inhibit many metabolic reprogramming pathways, such as glucose intake, aerobic glycolysis, glutamine metabolism, fatty acid anabolism, and nucleotide synthesis [35]. The relationship between the levels of total saturated fatty acids (SFAs) and oncogenesis is still a debatable subject in medicine. Considering that high levels of SFAs increase the risk of many cancers, a meta-analysis was performed to test this hypothesis. An evaluation of the articles revealed that SFA levels are not correlated with the incidence of GC [36]. However, lipid metabolism reprogramming affects normal immune cell function and plays a crucial role in the initiation and progression of cancer. Customized treatment can be directed by the lipid-metabolism-related gene (LRG)-based signature, which can independently predict the outcomes of GC patients [37].

### 3.2. Molecular and Genomic Characteristics

Although novel immunotherapies and targeted medications are being researched, not much has changed in this field. For the identification and treatment of precancerous lesions in GC, personalized therapy on the basis of clinical characteristics, pathology, and molecular typing is essential [38]. Macrophage-related genes may have enabled a new method of endothelial cell and macrophage communication in the TME. Additionally, the interaction between inflammation and angiogenesis may yield new targets for therapeutic intervention, opening new avenues for individualized treatment approaches [39]. Many studies have focused on the epigenetic control of N6-methyladenosine (m6A) modifications in tumors; nevertheless, the possible functions of genes associated with m6A regulators in the context of GC and the TME are still mostly unclear. To assess individual prognosis and treatment response, a scoring system was developed. Through unsupervised clustering of 56 m6A-regulator-related genes (all significantly correlated with GC prognosis), three unique m6A-regulator-related patterns were found. These patterns strongly correlated with immune-inflamed, immune-excluded, and immune-desert phenotypes according to TME categorization, and their TME features were highly consistent with various biological processes and clinical outcomes. In three different cohorts, a low score was associated with better responses to anti-PD-1/L1 and anti-cytotoxic T-lymphocyte-associated protein 4 (CTLA4) immunotherapy. The development of more customized treatment regimens and immunotherapy strategies may be facilitated by the established grading system [40]. Prognostic markers may be derived from genes associated with m6A regulation, as these genes may be linked to the initiation, growth, and progression of HER2-positive GC. Innovative medications that block methyltransferases such as 3 and other molecules by targeting m6A modifications have demonstrated remarkable anticancer effects. Therefore, by overcoming medication resistance, developing novel therapeutic alternatives using m6A can help patients with HER2-positive GC achieve better treatment outcomes. Overall, integrating m6A alterations into a clinical research system can advance targeted medicine, precision medicine, and treatment approaches for GC, particularly HER2-positive GC. However, more investigations into the molecular mechanism and related difficulties at the molecular level are necessary before these findings can be translated into clinical studies [41].

Compared with patients with early disease, patients with advanced GC exhibit greater frequencies of cumulative genetic events, such as increased rates of phosphatidylinositol-4,5-bisphosphate 3-kinase catalytic subunit alpha (PIK3CA) mutations, better detection of immunotherapy biomarkers, and mutations in the estrogen receptor 1 (ESR1) gene. In addition, genetic alterations such as PIK3CA, the KRAS proto-oncogene, GTPase (KRAS), and Erb-B2 receptor tyrosine kinase 2 (ERBB2), which act as somatic oncogenic drivers, are more frequently encountered [42]. Long noncoding RNAs (lncRNAs) are RNAs that do not encode proteins but instead transcribe more than 200 nucleotides. LncRNAs have multiple applications, including as scaffolds, guide molecules, signal molecules, and decoy molecules. To control gene expression in the cell nucleus, long noncoding RNAs can interact with transcriptional regulatory proteins, chromatin-modifying complexes, and DNA. MiRNAs are found throughout the cytoplasm; via miRNAs, other transcription products, and proteins, miRNAs also participate in the degradation of mRNAs and the control of translation. The function of RNF144A-AS1, commonly known as GRASLND, in controlling chondrogenic development in mesenchymal stem cells was first identified. However, RNF144A-AS1 also shows promise for the diagnosis and treatment of tumors [43]. Discovering target therapies and understanding illness mechanisms have been made possible by advances in molecular medicine. The management of diseases by doctors has changed with the increased accessibility of molecular tests such as proteomics, immunohistochemical analysis, and DNA and RNA sequencing. The Cancer Genome Atlas Program’s molecular categorization of GC separates GAC into four subgroups. However, these molecular subgroups seemingly have little to do with authorized immunotherapies or accessible targets. Clinical guidelines for cancer management in the era of molecular therapies rely on available actionable targets and approved therapies until a more trustworthy interpretation of the massive amount of data provided by the molecular classifications is presented [44].

The three most studied categories of noncoding RNAs are microRNAs, lncRNAs, and circular RNAs (circRNAs). Other noncoding RNAs (ncRNAs), such as lncRNAs and circRNAs, are increasingly recognized as significant contributors to human disease. We can also find studies on the relationship between microRNAs (miRNAs) and human cancers [45]. Researchers have built a competing endogenous RNA (ceRNA) network with 907 messenger RNAs (mRNAs) and 84 miRNAs. They identified two molecular subtypes using 26 genes from the TCGASTAD and GSE84437 dataset intersections. Subtype S2 was associated with a poor prognosis, a high immunological score, a low response to immunotherapy, and a low tumor mutational burden. Subtype S1 exhibited increased susceptibility to the AKT (protein kinase B, or PKB) inhibitors VIII, salubrinal, gemcitabine, vinorelbine, pyrimethamine, and sorafenib [46]. Several cancers, including GC, are controlled by circRNAs, which help CD8+ T cells undergo immunological evasion. Moreover, hsa_circ_0001479 promotes the growth and spread of GC [47]. Exosomal circRNAs have emerged as significant participants in GC. Many times, GC patients receive an advanced diagnosis. Consequently, screening and identification of circRNAs are crucial before initiating immunotherapy [48].

### 3.3. Possible Implications of Bacterial Pathogens

Certain bacteria have been shown to promote oncogenesis, whereas others seem to provide tumor protection. Research has indicated that alterations in the constitution and quantity of the microbiome may be linked to the emergence of some gastrointestinal malignancies, including those of the stomach, liver, colon, and esophagus [49]. GC patients frequently possess species of Fusobacterium, Clostridium, and Lactobacillus. To fully understand the role that *Fusobacterium nucleatum* plays in the development of GC, more research is needed [50]. A “hot” TME was indicated by greater densities of PD-L1+ cells and non-exhausted CD8+ T cells in *Helicobacter-pylori*-positive GC. According to transcriptome analysis, immunotherapy-sensitive GC and *H.-pylori*-positive GC have comparable molecular traits. Because *H. pylori* infection shapes a “hot” TME, it is advantageous for GC immunotherapy [51].

From an immunological point of view, more studies on immunotherapy in patients with *H.-pylori*-positive GC may be conducted in the coming years. In recent years, research has focused on exosomes, gamma-glutamyl transpeptidase (GGT), the gut microbiota, autophagy, immunotherapy, and epithelial–mesenchymal transition (EMT) [52]. One of the domains of interest is vaccine therapy. The excellent ratios of infiltrating CD8+/CD4+ T cells, decreased invasion of regulatory forkhead box P3 (FOXP3)+ Treg lymphocytes, increased apoptosis caused by cysteine-aspartic protease (Caspase) 9/Caspase 3 overexpression, and baculoviral inhibitor of apoptosis repeat-containing 5 (Survivin) downregulation suggest that *H. pylori* pIRES2-DsRed-Express-ureF DNA vaccines may have immunotherapeutic utility in individuals with advanced GC [53]. For the treatment of *H. pylori* infection, toll-like receptor 6 (TLR6) may be a promising immunotherapy target. Having a high affinity for bacteria and being responsible for the first step in activating the immune response influences the immunotolerance of *H. pylori*. Through TLR6/c-Jun N-terminal kinase (JNK) signaling, prolonged *H. pylori* infection decreases the sensitivity of TLR6 to bacterial components and controls the production of inflammatory cytokines in human gastric epithelial (GES)-1 cells. Blocking TLR6 may diminish bacterial colonization, thus preventing the oncogenesis process [54].

### 3.4. Possible Immunotherapeutic Resistance and Hyperprogressive Disease

According to a study conducted in C57BL/6J mice, ICI resistance in GC patients treated with Janus kinase inhibitors (JAKi) is overcome in cases of peritoneal spread. Increased tumor-specific CD8+ T-cell infiltration is how dual ICI therapy reduces tumor growth, and Janus kinase inhibitor (JAKi) supplementation enhances ICI resistance by changing the immunosuppressive TME [55]. In a different study, researchers examined the immune checkpoint expression properties of CD8+ T cells in GC and the immune checkpoint expression pattern (ICEP) mediating anti-PD-1 treatment resistance via single-cell RNA sequencing (scRNA-seq) and multivariate linear regression interaction models. On the basis of interaction research, the co-expression of PD-1 and NKG2A (also known as CD159) may have a more significant inhibitory effect on the proliferative capacity of CD8+ T cells than other immune checkpoints that are currently known. As defined by ICEP1 (CD8+ T cells co-expressing PD-1, CTLA-4, TIGIT (an immune receptor present on some T cells and natural killer cells), LAG-3 (lymphocyte-activation gene 3), or CD38) and ICEP2 (CD8+ T cells solely expressing NKG2A or co-expressing with other immune checkpoints), the co-expression analysis of PD-1 and NKG2A revealed a differential co-expression pattern, reflecting the cooccurrence pattern of PD-1 and the mutual exclusivity of NKG2A. These ICEP CD8+ T-cell subgroups indicated that distinct CD8+ T-cell development fates are regulated by activated B-cell factor-1 and runt-related transcription factor 3. ICEP2 CD8+ T cells are associated with anti-PD-1 therapy resistance in GC patients. The cause is the recruitment of legumain (LGMN)+ macrophages, which is carried out via the chemokine ligand 16 (CXCL16)-CXCR6 (a receptor for the chemokine CXCL16) signaling pathway [56]. Immunotherapeutic resistance is linked to V-domain immunoglobulin suppressor of T-cell activation (VISTA), an immunological checkpoint. TAMs reportedly express VISTA, which is associated with a poor prognosis and a low immunotherapy response. Therefore, GC therapy may be influenced by blocking VISTA. Its inhibition seems to reorganize TAM action, stimulate T-cell-mediated antitumour immunity, and enhance the effect of PD-1 inhibitors [57]. One important effect of immunotherapy that can increase the difficulty for clinicians is hyperprogressive disease (HPD). The incidence of HPD may differ depending on the tumor type and the definitions applied, which lends credence to the need for a standardized and enhanced process for evaluating HPD to make well-informed treatment decisions [58].

### 3.5. The Possibility of Assessing Prognosis and Immunotherapy Response through Biomarkers for the Benefit of Immunotherapy after Surgery

Survivability can be reliably predicted by a nomogram that combines clinical indicators and risk scores. This risk model can be a useful tool for predicting the prognosis and response to immunotherapy in patients with GC [59]. By combining clinical characteristics with risk signatures, a different nomogram was created that revealed excellent accuracy in forecasting 1-, 3-, and 5-year survival rates. Furthermore, there were differences in medication sensitivity between the high-risk and low-risk groups [60]. The low-risk group responded well to immune checkpoint blockade therapy and had a better prognosis. In addition, multiple RNA changes may have a significant impact on the prognosis of patients with GC [61]. Variations in GC prognosis may be associated with TME early 2-factor (E2F) patterns. For GC patients, the E2F score is a useful predictor of survival and responsiveness to treatment [62]. In GC, keratin 7 (KRT7) is strongly expressed. When KRT7 expression is reduced, GC cell motility and proliferation are markedly inhibited. The TME characteristics of GC patients were identified via models based on the PANoptosis signature, which can also be used to accurately predict the prognosis and immunological effectiveness of GCs [63]. Ferroptosis is a unique form of apoptosis that plays a role in oncogenesis by promoting tumor initiation, progression, and ultimately prognosis. Although the effects of ferroptosis-related genes (FRGs) on the TME have not been fully characterized, studies of the immune status and MSI-H status in a database of 44 FRGs revealed that, owing to their high mutational load, immunological activation, and MSI-H status, a low FRG score is associated with an increase in OS [64]. The importance of illness prognosis in patients with GC is unknown, as is the ratio of activated CD4+ T cells to Tregs that infiltrate the microenvironment. OS was compared between patients divided into two groups: those with a CD4 T cell/Treg ratio > 1 and those with a CD4 T cell/Treg ratio < 1. Survival analysis via the Kaplan–Meier method revealed that the group with a CD4+ T cell/Treg ratio < 1 had an unfavorable outcome. The immune-metabolism signature associated with the ratio of active CD4 T cells to Tregs may be used to assess the prognosis, TME, and treatment strategies for patients with GC following additional examination of low- and high-risk patient groups [65]. Thirteen differentially expressed (DE) ubiquitination-related genes (URGs) were used to construct a predictive signature that divided patients into two risk categories. Patients at higher risk had significantly shorter survival periods than those at lower risk. Medical professionals would thus have a trustworthy tool to enhance prognostic assessment and support clinical treatment decisions in the form of a URG-based genetic model that may accurately predict a patient’s prognosis and immunotherapy response [66]. The results revealed the potential effects of carcinogenesis-associated genes on the TME, pathological and clinical features, and prognosis of patients with GC. This signature, which is strongly correlated with the immune response against GC, may be a useful tool for predicting patient prognosis [67].

There was a significant correlation between lymph node metastasis and submucosal invasion (odds ratio 2.04, 95% CI: 1.58–2.63, I 2 = 88.7%; *p* < 0.001), vertical margin invasion (odds ratio 6.11, 95% CI: 1.94–19.23, I 2 = 0%; *p* < 0.001), lymphatic invasion (odds ratio 10.02, 95% CI: 7.57–13.27, I 2 = 92%; *p* < 0.000), and vascular invasion (odds ratio 7.11, 95% CI: 5.49–9.22, I 2 = 92%; *p* < 0.000). Therefore, while choosing criteria for surgical therapy, it is essential to carefully consider invasion of the lymph nodes, vascular system, submucosa, and positive vertical margin [68]. Patients who received neoadjuvant immunotherapy experienced two outcomes: pathological complete remission (24%; 95% CI: 19–28%) and substantial pathological remission (49%; 95% CI: 38–61%). For resectable gastric/esophageal junction cancers, neoadjuvant immunotherapy, particularly neoadjuvant dual immunotherapy combinations, represents a safe and beneficial option in the short term [69]. While prolonging the initial recurrence time of patients in the camrelizumab + nab-paclitaxel + S-1 (C-SAP) group, the combination of camrelizumab and neoadjuvant chemotherapy improved the rate of ypT0 (no evidence of tumor in the primary lesion, which represents the best response to preoperative therapy), ypN0 (sterilization of metastatic regional lymph nodes), and pathologic complete response in patients. In addition, it did not increase postoperative complications or side effects related to immunotherapy or chemotherapy [70].

Multiple malignancies, including GC, nasopharyngeal carcinoma, and lymphoma, are connected to the double-stranded DNA virus known as Epstein–Barr virus (EBV), a member of the Orthoherpesviridae family. Vaccines and immunotherapies are now key research objectives because there are no consistently effective treatments for human EBV infection [71]. EBV-associated GC (EBVaGC) could be a useful predictive marker for immunotherapy efficacy. According to the analysis of patients who underwent complete resection for localized GC at a Western Academic Institution, the incidence of surgically removed EBVaGC was 4%. The presence of robust tumor-infiltrating lymphocyte phenotypes and elevated expression of PD-L1 in EBVaGC may indicate positive responses to immunotherapy [72].

### 3.6. Other Factors Influencing Immunotherapy

Other contributing variables include comorbidities such as muscular atrophy, resistance to immunotherapy or chemotherapy (because of PD-L1 expression), and low expression of cancer genes [73]. Hypertension (29 patients), diabetes mellitus (23 patients), ischemic heart disease (5 patients), and other conditions (15 patients) were the primary comorbidities for patients with stage IV GC treated with HER2 inhibitors in combination with chemotherapy (18%), immune checkpoint inhibitors (15%), inhibitors of MET proto-oncogene, receptor tyrosine kinase (MET) or vascular endothelial growth factor (VEGFR)2 (5%), and first-line capecitabine/oxaliplatin (62%) [74]. Proteomics, metabolomics, and next-generation sequencing are some of the most recent advances in omics technologies that have shed light on possible genetic changes and biological processes in GC. Therefore, the treatment of GC as a single disease is not appropriate [75]. International standards are generally in agreement regarding the significance of gastric intestinal metaplasia (GIM) as a precancerous condition and the necessity of a risk-stratified strategy for endoscopic surveillance, as well as the eradication of *H. pylori* when present, despite diverse demographics and practices. Guidelines need to be harmonized regarding (1) which populations should be screened for GC via an index endoscopic procedure and GIM detection/staging; (2) objective metrics for high-quality endoscopy; (3) agreement on the necessity of histological staging; and (4) non-endoscopic interventions for the prevention of GC other than the removal of *H. pylori* alone [76]. On the basis of gene involvement and changes, patients in the low-socioeconomic-status group required targeted therapy or immunotherapy more frequently. The genomes of patients with stomach adenocarcinomas from various socioeconomic backgrounds differ significantly, which could indicate that these patients require different targeted therapies and immunotherapies [77].

As a result of these studies, it is essential to analyze a wide range of factors, from immunopathological characteristics to bacterial pathogens and genomic alterations. There is variability in patient presentation and many problems with the available resources. However, each university of medicine has specialized people in the departments of immunology, genetics, microbiology, internal medicine, and gastroenterology. With available human resources, the functionality of GC medical centers supported by prestigious medical universities remains at the state’s discretion, which allocates targeted financial resources.

## 4. Conclusions

Immunotherapy for GC has been extensively studied because of the high fatality rate of this disorder. Numerous fields of immunotherapy are the subject of research. Understanding and improving the success rate of immunotherapy requires an in-depth investigation of all these aspects of immunotherapy. For this reason, we propose that before immunotherapy is started for patients with GC, a score ranging from 1–2 points should be assigned for the following patient aspects: 1. immunopathological characteristics, such as CD8+ T cells and peripheral CD4+ T cells; 2. molecular and genomic characteristics, such as N6-methyladenosine (m6A), HER2, and circRNA levels; 3. possible implications of bacterial pathogens, such as Fusobacterium, Clostridium, Lactobacillus, and *H. pylori*; 4. possible immunotherapeutic resistance and hyperprogressive disease, such as the VISTA gene and hyperprogressive disease; and 5. the possibility of assessing prognosis and immunotherapy responses through biomarkers for the benefit of immunotherapy after surgery, such as multiple RNA changes, E2F patterns, KRT7, ferroptosis-related genes, the ratio of active CD4+ T cells to Tregs, and ubiquitination-related genes. Patients who receive the highest possible score on the proposed protocol may be eligible for immunotherapy as soon as they are diagnosed. Consequently, the physician’s first course of action may be immunotherapy.

## Figures and Tables

**Figure 1 pharmaceuticals-17-01121-f001:**
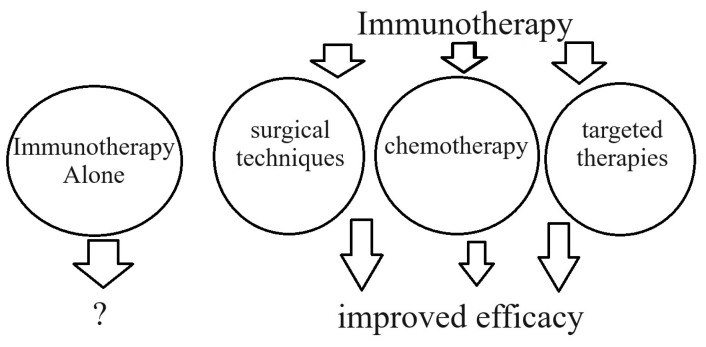
Immunotherapy alone or in combination with other therapies as a first-line treatment for GC?

**Table 1 pharmaceuticals-17-01121-t001:** Major developments in GC immunotherapy.

What Is New in Immunotherapy for GC?	References
- ICIS, tumor vaccines, adoptive immunotherapy, and nonspecific immunomodulators hold great potential, particularly for incurable or metastatic conditions.	[5]
- An increasing number of cancers, including GC, have been approved for treatment with ICIs.	[6]
- A possible biomarker for forecasting the ICI response in GC might be the PTCH1 mutation.	[7]
- As a first-line treatment for advanced GC, apatinib, in conjunction with chemotherapy and immunotherapy, has significant antitumour effectiveness and is well tolerated by patients.	[8]
- In second-line therapy for advanced GC or esophagogastric junction cancer, toripalimab and apatinib produced acceptable toxicity but did not enhance clinical outcomes over chemotherapy treatment.	[9]
- Immunotherapy plus chemotherapy may also be beneficial for certain subgroups within the low PD-L1-expressing population, such as those with non-diffuse-type tumors and no peritoneal metastases.	[10]
- When used with nivolumab, oligo-fractionated irradiation can have an immune-modulating impact and potentially disseminate antigens in a subset of patients with GC.	[11]
- The percentage of patients who benefit from anti-PD-1 treatments may be increased by strategies that promote the development of antitumour immune hubs.	[12]
- PD-1 inhibitor-responsive patients can be identified primarily by their baseline CEA level, which can be a predictive biomarker.	[13]
- In HER2-positive cancer treatment, combinatorial immunotherapeutic methods incorporating PD-1/PD-L1 inhibitors show potential.	[14]
- The first-line treatment for the HER2 GC subtype is combined immunotherapy plus anti-HER2 drugs.	[15]
- The first ‘targeted’ treatment specifically for GC is targeting Claudin-18.2.	[16]
- CLDN18.2 is expressed on the outer cell membrane; mAbs can bind to CLDN18.2.	[17]
- The KEYNOTE-859 and SPOTLIGHT trials, which use ICIs and zolbetuximab to target CLDN18.2, respectively, address an unmet need for additional targeted therapies for patients with previously untreated, HER2-negative, unresectable, or metastatic GC.	[18]
- PFS, OS, and ORR did not significantly differ between checkpoint inhibitors or between immunotherapy and anti-claudin-18.2-targeted medicines when used as the first line of treatment for HER2-negative, incurable, or metastatic gastric malignancies, according to a network meta-analysis.	[19]
- An essential component of cancer immunotherapy is the ligand’s interaction with the receptor, which determines whether to stimulate or inhibit immune cell activity.	[20]
- Immunotherapy has demonstrated high efficacy with manageable toxicity levels compared to standard medicines.	[21]
- A complete objective and ctDNA response in salvage therapy for metastatic GC can be safely provided by CAR T cells targeted for Claudin 18.2.	[22]
- The antitumour response mediated by CAR-T cells was enhanced in GC when GLIPR1 knockdown was paired with CAR structural design.	[23]
- Before using ICIs to treat GC, it is critical to identify MSI.	[24]
- Immunotherapy does not work for some MSI-H/dMMR GC patients	[25]
- Anti-PD-L1 treatment can predict the development of irAE since anti-PD-L1-mediated lysosomal degradation stimulates the generation of soluble PD-L1.	[26]
- Tislelizumab was well tolerated and showed a significant improvement in ORR in patients with previously treated MSI-H/dMMR cancers that were either locally progressed, unresectable, or metastatic.	[27]
- Preclinical research supports IBI315 as a viable bispecific antibody-based immunotherapy strategy for HER2-positive GC, expanding the treatment options available to this patient population	[28]
- A CLDN18.2×4-1BB BsAb called givastomig/ABL111 may be able to treat GC patients with a broad range of CLDN18.2 expression	[29]
- The continued study of BsAb-based immunotherapy for solid tumors may soon yield promising results for clinical use.	[30]

## Data Availability

Data sharing is not applicable.

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
