# Peer review of "A New Medical Evaluation for Gastric Cancer Patients to Increase the Success Rate of Immunotherapy: A 2024 Update"

_pharmaceuticals, 2024, doi:10.3390/ph17091121_

Round 1

Reviewer 1 Report

Comments and Suggestions for Authors

The manuscript by Samasca et al. is a review article that summarizes recent advances and provides an update on the most important developments in gastric cancer with a special focus on immunotherapy research. Furthermore, based on this update, the authors propose a new medical evaluation before starting immunotherapy in gastric cancer. Thus, this review is interesting and timely. However, in my opinion, the presentation could and should be improved.

The title is unclear and incomplete: "A new medical evaluation" of what?

I think the authors should add figures/tables to illustrate the text.

Although I do not feel good enough to judge the quality of the English in this paper, I can see many errors that should be corrected. 

Author Response

The title is unclear and incomplete: "A new medical evaluation" of what?

  • We thank you for your suggestions. We have changed the title to: “A new medical evaluation for gastric cancer patients to increase the success rate of immunotherapy: a 2024 update”

I think the authors should add figures/tables to illustrate the text.

  • We thank you for your suggestions. We have inserted in the text Figure 1 and Table 1.

Although I do not feel good enough to judge the quality of the English in this paper, I can see many errors that should be corrected. 

We thank you for your suggestions.  We have revised the whole article and corrected the English language.

Reviewer 2 Report

Comments and Suggestions for Authors

The manuscript entitled “A new medical evaluation to increase the success rate of immunotherapy in gastric cancer: a 2024 update” by Samasca et al. is a comprehensive review focused on understanding the progress as well as challenges of immunotherapy in the context of gastric cancer (GC). The authors introduced immunotherapy and discussed the progress of GC immunotherapy in detail. This was followed by a vivid description of immunotherapy characteristics including molecular and genomic characteristics, implications of infection with bacterial pathogen etc. The authors concluded that the success rate of immunotherapy can be increased by an evaluation of certain defined characteristics or parameters before starting the immunotherapy.

I thoroughly enjoyed reading the review article and strongly believe that the new evaluation proposal might provide new insights about immunotherapy and that can be extended in other types of metastases beside GC.

The only suggestion/comment I have is to include non-coding RNAs other than lncRNAs that are also implicated in GC that the authors might have missed.

Otherwise, I recommend that the review article can be accepted for publication.

Author Response

We thank you for your suggestions. We have introduced a paragraph about the importance of circRNAs in CG.

The three most studied categories of non-coding RNAs are microRNAs, long non-coding RNAs (lncRNAs), and circular RNAs (circRNAs). Other non-coding RNAs (ncRNAs), such as long non-coding RNAs (lncRNAs) and circular RNAs (circRNAs), are increasingly becoming recognized as significant contributors to human disease. We can also find studies on the relationship between microRNAs (miRNAs) and human cancers [43]. The researchers built a competing endogenous RNA (ceRNA) network with 907 messenger RNAs (mRNAs) and 84 miRNAs. They identified two molecular subtypes using 26 genes from the TCGASTAD and GSE84437 dataset intersections. Subtype S2 was associated with a poor prognosis, a higher immunological score, a lower response to immunotherapy, and a lower tumor mutational burden. Subtype S1 exhibited increased susceptibility to AKT (protein kinase B, or PKB) inhibitor VIII, salubrinal, gemcitabine, vinorelbine, pyrimethamine, and sorafenib [44]. Several cancers, including GC, are controlled by circRNAs, which help CD8+T cells undergo immunological evasion. hsa_circ_0001479 promotes the growth and spread of GC [45]. Exosomal circRNAs have emerged as significant participants in GC. Many times, GC patients receive an advanced diagnosis. Consequently, screening and identification of circRNAs are crucial before initiating immunotherapy [46].

Reviewer 3 Report

Comments and Suggestions for Authors

PFA

Author Response

  • The review comprehensively covers the relevant literature on the topic: “Increase the success rate of immunotherapy in gastric cancer”. The idea of evaluating the GC patients for immunotherapy based on the proposed aspects is very interesting. The review provides a balanced overview of using these divisions to enhance the success of immunotherapy in GC patients.

We thank you for your comments.

  • More literature should be included in the section on Cancer immunotherapy based on bispecific antibodies, showing what research has been done and what can be the future possibilities.

We thank you for your comments. We have added the following paragraph:

The development and improvement of bispecific antibody (BsAb) structures, as well as their combination with other therapeutic modalities to increase their efficacy and overcome resistance, are the main topics of current research. What research has been done? Using patient-derived xenografts and organoids, a recombinant fully human IgG1 bispecific antibody, IBI315, was created, and its antitumor efficacy, as well as the underlying mechanism, were studied in vitro and in vivo in mouse tumor models reconstituted with human immune cells. Tumor immune cell recruitment and activation are also induced by IBI315 therapy. Tumor cells undergo pyroptosis mediated by gasdermin B (GSDMB), which is triggered by IBI315 and results in T cell recruitment and activation. [28]. A new CLDN18, known by other names such as TJ-CD4B or TJ033721, the 2×4-1BB bispecific antibody 'givastomig' or 'ABL111' was created to trigger 4-1BB signaling in a way that was dependent on CLDN18.2 interaction. In patients with GC (n = 60), 4-1BB+ T cells and CLDN18.2+ tumor cells were found to coexist. Givastomig/ABL111 could only elicit 4-1BB activation in vitro in the setting of CLDN18.2 binding and bind with high affinity to cell lines expressing different amounts of CLDN18.2. There was a strong correlation between the degree of T-cell activation induced by givastomig/ABL111 therapy and the CLDN18.2 expression level in tumor cells obtained from the GC patient xenograft model [29]. Thus, tumor models were used in the current BsAb investigation. What are the future possibilities? Cancer immunotherapy based on BsAbs has opened up new treatment options for many types of cancer. In the future, there is hope that the current investigation of BsAb-based immunotherapy for solid tumors may provide promising clinical application prospects [30].

  • The limitations of immunotherapy are mentioned well in section 3.4, but a little more elaboration is needed including the aspect of cell storm in immunotherapy.

We thank you for your comments. We have added the following paragraph:

According to a study conducted in C57BL/6 J mice, ICIs resistance in GC patients treated with Janus kinase inhibitors (JAKi) is overcome in cases of peritoneal spread. Increased tumor-specific CD8+ T cell infiltration is how dual ICIs therapy reduces tumor growth, and Janus kinase inhibitor (JAKi) supplementation enhances ICIs resistance by changing the immunosuppressive TME [55]. In a different study, the researchers examined the immune checkpoint expression properties of CD8+ T cells in GC and the immune checkpoint expression pattern (ICEP) mediating anti-PD-1 treatment resistance by utilizing single-cell RNA sequencing (scRNA-seq) and multivariate linear regression interaction models. Based on interaction research, co-expression of PD-1 and NKG2A (also known as CD159) may have a more significant inhibitory effect on CD8+ T cell proliferative capacity than other immune checkpoints that are currently known. As defined by ICEP1 (CD8+ T cells co-expressing PD-1, CTLA-4, TIGIT (an immune receptor present on some T cells and natural killer cells), LAG-3 (lymphocyte-activation gene 3), or CD38) and ICEP2 (CD8+ T cells solely expressing NKG2A or co-expressing with other immune checkpoints), the co-expression analysis of PD-1 and NKG2A revealed a differential co-expression pattern, reflecting the co-occurrence pattern of PD-1 and the mutual exclusivity of NKG2A. These ICEP CD8+ T cell subgroups indicated distinct CD8+ T cell development fates regulated by activated B-cell factor-1 and runt-related transcription factor 3. ICEP2 CD8+ T cells are associated with anti-PD-1 therapy resistance in GC patients. The cause is the recruitment of legumain(LGMN)+ macrophages, carried out by the chemokine ligand 16 (CXCL16)-CXCR6 (a receptor for the chemokine CXCL16) signaling pathway [56].

  • The overall flow of the article is good and well-structured providing a clear reading path to the readers.

We thank you for your comments.

  • The recommendation to consider a wide range of factors, from immunopathological characteristics to bacterial pathogens and genomic alterations, is ambitious. While comprehensive, this scope may be too broad for practical clinical application, especially given the variability in patient presentation and available resources.

We thank you for your comments. We have added the following before Conclusions:

As a result of the mentioned studies, it is essential to analyse a wide range of factors, from immunopathological characteristics to bacterial pathogens and genomic alterations. There is variability in patient presentation and many problems with available resources. However, each university of medicine has specialized people in the departments of immunology, genetics, microbiology, internal medicine, and gastroenterology. With the available human resources, the functionality of GC medical centers supported by prestigious medical universities remains at the state's discretion, which allocates the targeted financial resources.

  • A few other factors such as patient comorbidities, prior treatments, heterogeneity of GC, or overall health status and their effect on the provided immunotherapy can also be discussed to provide a better understanding of the effect of immunotherapy on GC patients.

We thank you for your comments. We have added the following paragraph:

3.6. Other factors influencing immunotherapy

Other contributing variables include comorbidities such as muscular atrophy, resistance to immunotherapy or chemotherapy (because of PD-L1 expression), and low expression of the cancer gene [73]. Hypertension (29 patients), diabetes mellitus (23 patients), ischemic heart disease (5 patients), and other conditions (15 patients) were the primary comorbidities for patients with stage IV GC treated with HER2 inhibitors in combination with chemotherapy (18%), immune checkpoint inhibitors (15%), inhibitors of MET Proto-Oncogene, Receptor Tyrosine Kinase (MET) or Vascular Endothelial Growth Factor (VEGFR)2 (5%), and first-line capecitabine/oxaliplatin (62%) [74]. Proteomics, metabolomics, and next-generation sequencing are some of the most recent advances in omics technologies that have shed light on possible genetic changes and biological processes in GC. Therefore, treatment of GC as a single disease is not appropriate [75]. International standards are generally in agreement regarding the significance of gastric intestinal metaplasia (GIM) as a precancerous condition and the necessity of a risk-stratified strategy for endoscopic surveillance, as well as the eradication of H. pylori when present, despite diverse demographics and practices. Guidelines need to be harmonized about: (1) which populations should be screened for GC using an index endoscopic procedure and GIM detection/staging; (2) objective metrics for high-quality endoscopy; (3) agreement on the necessity of histological staging; and (4) non-endoscopic interventions for the prevention of GC other than the removal of H. pylori alone [76]. Based on gene involvement and changes, patients in the low socioeconomic group required targeted therapy or immunotherapy more frequently. Patients with stomach adenocarcinomas from varying socioeconomic backgrounds differ significantly in their genomes, which could mean that they require different targeted therapies and immunotherapies [77].

  • The conclusion does not specify how the proposed scoring system would impact clinical decision-making or how it would improve current practices. It lacks a discussion on how the proposed method would be validated and integrated into existing treatment protocols. Also, it seems the same as abstract. It should be rewritten.

We thank you for your comments. We have added the following paragraph in Conclusions

Patients who receive the highest possible score on the proposed protocol may be eligible for immunotherapy as soon as they are diagnosed. Consequently, the physician's first course of action may be immunotherapy.

We thank you for your comments. We have added the following paragraph in Abstract

The proposed scoring system would also help in the diagnosis of GC. With all the advances in genetics, immunology, and microbiology, it would be time for the diagnosis of GC to be improved, not changed! Currently, patients diagnosed with GC have surgical resection as the only permanent solution. Patients who meet the maximum score from the presented proposal could be eligible immediately after diagnosis for immunotherapy. Therefore, immunotherapy could be a first-line option for the clinician.